# AQP4, Astrogenesis, and Hydrocephalus: A New Neurological Perspective

**DOI:** 10.3390/ijms231810438

**Published:** 2022-09-09

**Authors:** Leandro Castañeyra-Ruiz, Ibrahim González-Marrero, Luis G. Hernández-Abad, Seunghyun Lee, Agustín Castañeyra-Perdomo, Michael Muhonen

**Affiliations:** 1CHOC Children’s Research Institute, 1201 W, La Veta Avenue, Orange, CA 92868, USA; 2Departamento de Ciencias Médicas Basicas, Anatomía, Facultad de Medicina, Universidad de La Laguna, Ofra s/n, 38071 La Laguna, Spain; 3Instituto de Investigación y Ciencias de Puerto del Rosario, 35600 Puerto del Rosario, Spain; 4Neurosurgery Department at CHOC Children’s Hospital, 505 S Main St., Orange, CA 92868, USA

**Keywords:** AQP4, astrogenesis, pediatric hydrocephalus, reactive astrogliosis, premature gliogenesis

## Abstract

Aquaporin 4 (AQP4) is a cerebral glial marker that labels ependymal cells and astrocytes’ endfeet and is the main water channel responsible for the parenchymal fluid balance. However, in brain development, AQP4 is a marker of glial stem cells and plays a crucial role in the pathophysiology of pediatric hydrocephalus. Gliogenesis characterization has been hampered by a lack of biomarkers for precursor and intermediate stages and a deeper understanding of hydrocephalus etiology is needed. This manuscript is a focused review of the current research landscape on AQP4 as a possible biomarker for gliogenesis and its influence in pediatric hydrocephalus, emphasizing reactive astrogliosis. The goal is to understand brain development under hydrocephalic and normal physiologic conditions.

## 1. Introduction

Although 90% of brain tissue is composed of glial cells, most cellular brain studies have focused on neuronal physiology; consequently, the glial cells have been relegated to a supportive position. Astrocytes are the most common glial cells in the CNS, accounting for 20 to 40% of total brain cells [1] and play a fundamental role in neural development, neural circuit function, neurotransmission, blood–brain barrier creation, and neural metabolic support. Astrocytes support the neurovascular systems by connecting neurons and endothelial cells, maintaining brain homeostasis, controlling water, amino acid, and neurotransmitter intake, and monitoring the local activity of synaptic circuits [2,3,4,5].

Since Ramon y Cajal proposed the neuron theory in the early 20th century [6], hundreds of neuron types and functions have been identified [7,8]. However, astrocytes are still considered a homogeneous population, only classified as protoplasmic or fibrous [9]. Astrocytes’ morphological and functional diversity, including their critical role in governing neuronal activity, is well-accepted. Therefore, defining astrogliogenesis to identify astrocytes’ functional and anatomical heterogeneity is crucial to understanding brain physiology [5,10].

Astrogenesis characterization is impeded by a lack of precursor and intermediate stages markers. Furthermore, astrocytes’ plasticity allows proliferative capabilities after being differentiated, complicating their identification [5,10]. Recently, our group has proposed the water channel AQP4 as a possible biomarker of gliogenesis, and its variations were characterized under control and pathological conditions. In-depth systematic reviews have been published, focusing on neuromyelitis optica [11] and gliomas [12,13]. This review focuses on AQP4 expression as a possible biomarker of astrogenesis development and its hypothetical relationship with pathology causing pediatric hydrocephalus.

### 1.1. AQP4: A Possible Astrogenesis Marker

Even though specific markers do not exist for the different stages of astrogenesis, it has been suggested that astrogenesis progresses trough at least four cellular stages: a radial glial cell (RGC), an intermediate progenitor cell, a maturing postnatal astrocyte, and an adult astrocyte. RGCs are fundamental in early brain development, serving as a scaffold for intermediate progenitors and neuro-glial precursors. After the cessation of neurogenesis, the RGCs become gliogenic (gliogenic switch) and the intermediate progenitors and astrocyte precursors migrate away from the germinal areas to differentiate into astrocytes [5,10,14].

AQP4 is a water channel linked to a glial lineage in the brain since it is expressed in astrocytes (mainly in the endfeet) and the ependymal cells [15,16]. AQP4 is expressed in neural stem cells (NSC) and their glial progeny [17] and modulates the proliferation, survival, migration, and neuronal differentiation of adult NSCs [18,19,20]. Interestingly, in adult lesser hedgehog tenrec (Echinops *telfairi*), immature radial glial cells persist through adulthood without expressing AQP4 [21]. In zebrafish, another form of astroglial cells continues through adulthood as RGCs expressing AQP4 [22]. In primates, cortical interlaminar astrocytes seem to have an RGC origin since they express typical RGC markers and adult astrocytes markers such as AQP4 [23]. All the animal findings support that AQP4 is associated with an astroglial lineage that may remain undifferentiated as RGCs in adulthood. In utero, AQP4 labels RGCs committed to the astrocyte lineage in humans [24,25] and mice [26]. It has been proposed that unpolarized AQP4-positive cells in the brain show proliferative and regenerative properties as neural stem cells [27]. In mice, the unpolarized expression of AQP4 in RGCs is not detected at the end of in utero life (E16), and the expression of AQP4 is restricted to the astroglial endfeet postnatally (polarized), at P1–3 [26] However, in humans, the AQP4-positive RGCs are present at the beginning of the second trimester of the pregnancy (13–14 postconceptional weeks, PCW), and the polarized expression of AQP4 forming the neurovascular unit is detected at the beginning of the third trimester (25 PCW) [24,25]. Therefore, the maturation of the expression of AQP4 seems to be delayed in mice compared to humans. Our group reported that unpolarized expression of AQP4 is explicitly found in a subpopulation of RGCs that do not progress radially toward the cortical plate but curve to follow white matter tracks, thus serving as a scaffold for glioblasts to develop and populate the white matter tracks, which in turn, provides functionality and maturity to the axons. Thus, these AQP4-positive RGCs were coined as glial stem cells (GSCs) because the projections are not radial and are committed to astrocytes [25].

Interestingly, GSCs follow a temporospatial expression pattern that may indicate the end of neurogenesis and the beginning of gliogenesis [24,25]. GSCs’ first detection is in the glioepithelium of the fimbria of the archicortex and progresses toward fibrous tracts, such as the corpus callosum (CC), the fornix, and the internal capsule in medial areas of the brain. From 25 PCW onward, AQP4 is expressed throughout the neocortex, mainly in the intermediate zone or subplate, giving rise to the cerebral cortex’s white matter. AQP4 is expressed primarily on the main fibrous tracts of the isocortex, possibly to provide homeostasis to facilitate neural impulse transmission to incipient fiber tracts. In general terms, AQP4-positive GSC expression progresses from medial to lateral, starting in the dorsal and ventral hippocampus archicortex, followed by the CC and the ganglionic eminences (transitory structures that contribute to the development of first neurogenic, later gliogenic, and ultimately degenerate), and finally, terminating in the intermediate zone of the resting brain. AQP4-dependent maturity progresses from medial to polar, initially in the perisylvian regions and finally in the occipital and prefrontal zones [24,25,28] (see Figure 1). This correlates with CC maturation. According to classic neuroanatomic studies and recent human embryology neuroimaging, the colossal connections begin centrally in the hippocampal primordium and progress bidirectionally both anteriorly and posteriorly [29,30,31,32,33], with more prominent anterior growth [34,35]. The expression pattern of AQP4-dependent GSCs could define a developmental pathway for cortical neuron functionality, with the archicortex (primitive cortex) acquiring functionality in early gestation, while the frontal and occipital poles acquire functionality at the end of gestation. Therefore, there is an ontogenic logic in which occipital and prefrontal areas achieve functionality and maturity late in brain development since they are responsible for the vision and complex behaviors such as the expression of personality, respectively.

In summary, unpolarized expression of AQP4 is found in proliferative cells with different morphologies: GSCs (with a long projection that is used for other cells to migrate and populate the white matter tracks), and intermediate progenitor cells (without a long projection and oval shape). We hypothesize that the expression of AQP4 in these progenitor cells indicates a gliogenic switch that represents the early stages of astrogenesis. Finally, when AQP4 expression is polarized (astrocytes’ endfeet), this indicates astrocyte maturity as a final step of the astrogenesis. Currently, three specific markers are accepted to identify astrocyte precursors: GLAST, FABP7/BLBP/, and FGFR3 [36,37,38,39]. GLAST is a glutamate transporter active in astrocytes, and its expression initiates with the gliogenic switch indicating specifically astrogenesis precursors. However, FABP7/BLBP/and FGFR3 are also expressed during neurogenic stages, making them unspecific markers [14]. In humans, AQP4 initiates its expression at the gliogenic switch, labeling different astrocyte precursors [24,25]. Thus, this water channel shows a similar expression pattern to GLAST and could be a relevant marker of astrogenesis. Further studies in experimental models should be conducted to confirm this.

### 1.2. AQP4 in Pediatric Hydrocephalus: Neurodevelopmental Implications

Hydrocephalus is an abnormal build-up of cerebrospinal fluid associated with distension of the ventricular system due to impairments in CSF circulation [40,41]. Inflammatory response plays a fundamental role in acquired conditions such as postinfectious and posthemorrhagic hydrocephalus [42,43,44,45,46]. Reactive astrogliosis (one of the primary expressions of neuroinflammation) is associated with both congenital and acquired hydrocephalus [45,47,48,49,50,51,52,53,54]. Historically, the neurodevelopmental consequences of pediatric hydrocephalus are attributed to parenchymal stretch and periventricular white matter injury secondary to ventricular enlargement and elevated intracranial pressure [55]. Recent evidence supports that the alterations in brain development (neuro-gliogenesis) are responsible for the etiology of some congenital hydrocephalus cases. Thus, genetic disruption of the neuro-gliogenesis in early brain development without CSF dynamic impairment could be a primary pathological factor in patients with congenital hydrocephalus [56]. In addition, GRC and neural progenitor cell loss due to VZ disruption are associated with abnormalities in neurogenesis such as abnormal neuroblast migration as intrinsic mechanisms of congenital [57,58,59] and posthemorrhagic hydrocephalus [60,61]. AQPs have been highly studied in brain development under control [24,28,62] and hydrocephalic conditions [63,64,65,66,67,68,69,70,71]. It has been proposed that the AQP variations are associated with hydrocephalus-compensatory mechanisms to decrease the production and increase the absorption of CSF [66,67,68,69,70,71]. AQPs have been proposed as possible CSF biomarkers for the diagnosis and prognosis of hydrocephalus [63,64,65,66].

Interestingly, AQP4 modulates the neurogenesis associated with neuroinflammation [20] and the glial proliferation in astrocyte cultures [72]. Thus, variations of the expression of AQP4 in the early stages of fetal-onset hydrocephalus may induce neurodevelopmental disorders associated with decreased neurogenesis. Our group reported an early differentiation of AQP4 positive GSCs into reactive astrocytes, implying impaired scaffold functionality in congenital hydrocephalus with spina bifida [25]. The GSCs’ early differentiation into reactive astrocytes could be considered a complementary neurodevelopmental disorder since the GSCs lose their projection, and the glioblasts are no longer migrating toward the white matter tracks. A similar mechanism of premature progenitor differentiation into astrocytes has been recently confirmed in a rat model with spina bifida [73]. This early differentiation into astrocytes could explain the characteristic white matter alterations found in pediatric hydrocephalus [74,75,76]. In spina bifida patients, the CC shows partial agenesis in 65% of the cases and hypoplasia in 35%. In obstructive hydrocephalus due to mesencephalic aqueduct obstruction, 35% of the cases showed partial agenesis and 50% hypoplasia of the CC. Finally, 100% of patients with posthemorrhagic hydrocephalus showed CC hypoplasia [76]. Therefore, this premature differentiation of GSCs into reactive astrocytes may contribute to the neuro-gliogenesis disorders in pediatric hydrocephalus (See Figure 2).

In experiments with neural stem cell cultures from mice that differentiate into ependymal cells, a similar mechanism of premature gliogenesis was proposed under posthemorrhagic conditions [77,78]. In these experiments, neural stem cells committed to becoming ependymal cells differentiated into reactive astrocytes changing the final fate of the cells. These findings support a unified mechanism of neurodevelopmental disorder associated with early astrocyte differentiation. This differentiation in congenital and acquired hydrocephalus conditions may result in primary white matter alterations.

## 2. Future Work

According to the bibliographic evidence reported in this manuscript, AQP4 is a specific gliogenesis marker that labels glial stem cells associated with white matter tracks. It also affects the degree of glial development as more polarization is associated with maturity (endfeet in neurovascular unit), while no polarization is associated with undifferentiated cells (GSCs). The glial stem cells and neuroblasts seem to show reactivity under hydrocephalic conditions. This reactivity triggers an early differentiation into astrocytes affecting cell migration and normal neural development, possibly explaining primary white matter alterations such as agenesis or hypoplasia of the corpus callosum. Current treatments cannot improve this alteration since it is related to primary neurological development disorders and not to intracranial pressure. Therefore, complementary treatments should not only focus on treating intracranial hypertension but also on preventing cellular neurodevelopmental impairments. Since the inflammatory component plays a fundamental role in reactive astrogliosis, several anti-inflammatory drugs have been proposed to decrease the glial response under hydrocephalic conditions to improve linked cellular alterations. For example, decorin, a proteoglycan that plays a fundamental role in immune regulation and inflammatory diseases [79], reduces white matter-dependent alterations and cytopathology associated with juvenile hydrocephalus [80,81,82].

## 3. Search Strategy and Selection Criteria

Searches on Pubmed and Google Scholar from 1950 to June 2022 and pertinent papers were used to find references for this review. The searching terms were “hydrocephalus and gliogenesis”, “reactive astrogliosis and white matter and hydrocephalus”, “AQP4 and gliogenesis”, “AQP4 and brain development”, and “AQP4 and neurogenesis”. There were no language restrictions.

## 4. Limitations

This review did not strictly adhere to PRISMA guidelines [83]. This review focused primarily on AQP4, gliogenesis, and hydrocephalus, other pathologies associated with AQP4 such as neuromyelitis optica or gliomas were not covered.

## Figures and Tables

**Figure 1 ijms-23-10438-f001:**
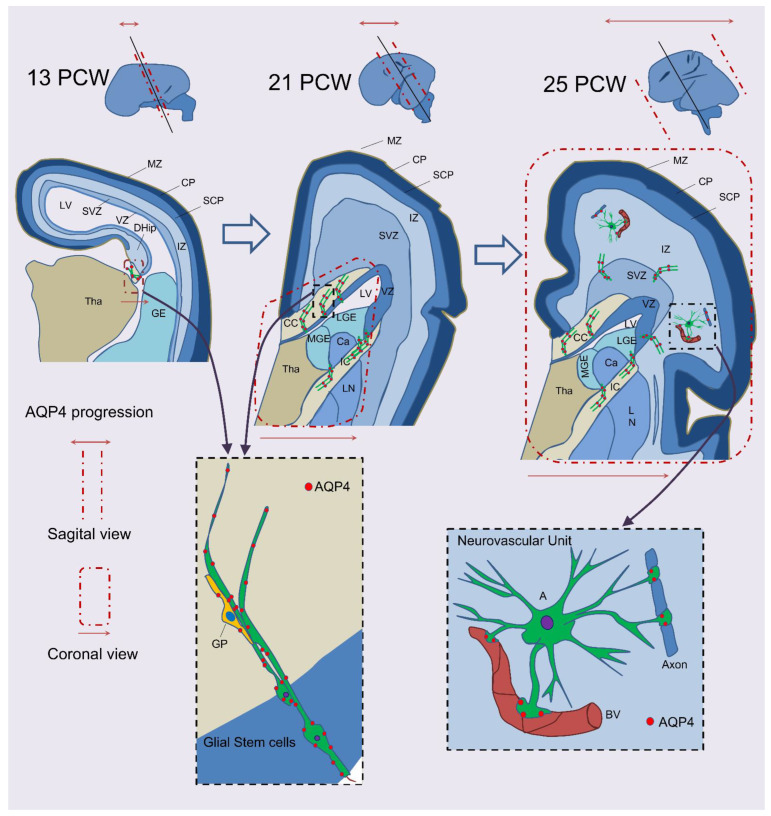
Schematic representation of the expression of AQP4 in brain development. In the telencephalon, AQP4 expression starts in GSCs and GBs and progresses from the para hippocampal VZ toward the glioepithelium of the fimbria of the dorsal hippocampus at 12–13 PCW. At 21 PCW, AQP4 has advanced medial to lateral in the coronal view, and medial to polar in the sagittal view. At this point, AQP4 is expressed in the ventricular zone adjacent to the medial portion of the CC, where the GSCs project their cellular process toward CC fibers. GSCs are also found in the LGE and the lenticular nucleus projecting toward the internal capsule. At 25 PCW, the AQP4 is patent cortically, and GSC processes are found projecting from the SVZ to the IZ. Polarized AQP4 expression is located in the astrocyte’s endfeet as a part of the neurovascular unit at the IZ, indicating maturity and functionality. LV, lateral ventricle; MZ, marginal zone; CP, cortical plate; SCP, subcortical plate; IZ, intermediate zone; SVZ, subventricular zone; VZ, ventricular zone; CC, corpus callosum; IC, internal capsule; DHip, dorsal hippocampus; Tha, thalamus; GE ganglionic eminence; LGE, lateral ganglionic eminence; MGE, medial ganglionic eminence; Ca, caudate; LN, lenticular nucleus; GP, glial progenitor; A, astrocyte; BV, blood vessel; AQP4, aquaporin 4.

**Figure 2 ijms-23-10438-f002:**
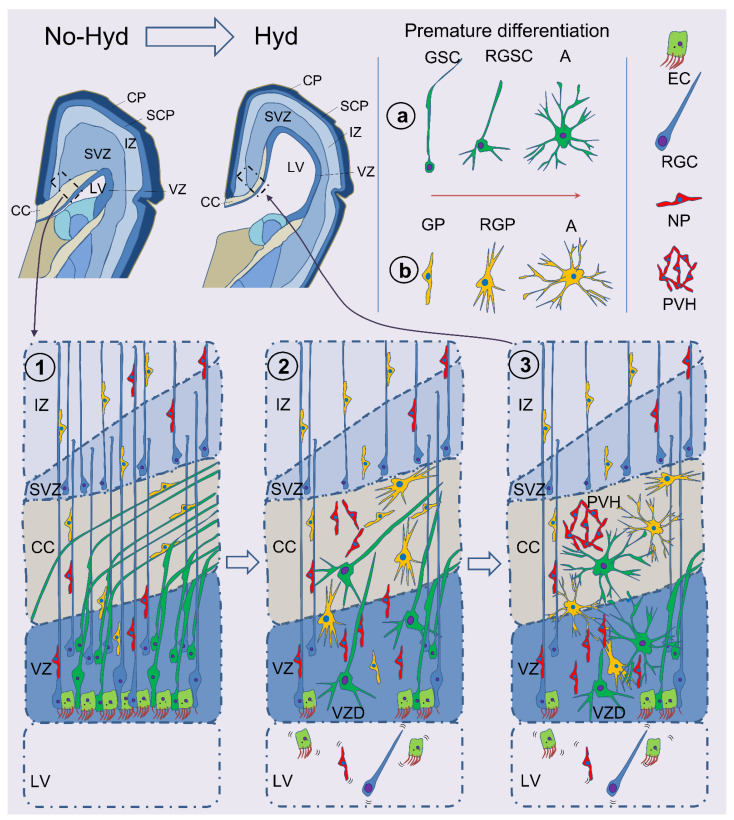
Schematic representation of astrocyte’s premature differentiation in hydrocephalus. In non-hydrocephalic conditions, after mid-gestation at the periventricular white matter areas, the ventricular zone is mainly composed of ECs, RGCs, GSCs, GPs, and NPs (1). When hydrocephalus pathology develops, a ventricular lining disruption occurs, associated with reactive astrogliosis (2). This representation proposes that white matter-associated GSCs and GPs become activated under hydrocephalic conditions suffering a premature differentiation into astrocytes (a,b). This early GSC differentiation into astrocytes impairs its normal function as a scaffold to guide cells into the white matter, implying a lack of cell migration resulting in the characteristic hypoplasia or dysgenesis of the white matter tracks. In turn, the incapability of NPs to migrate triggers periventricular heterotopias. Finally, a premature differentiation of GPs into astrocytes as a reaction to an inflammatory response may alter the final fate of the cells and affect their migration (3). LV, lateral ventricle; MZ, marginal zone; CP, cortical plate; SCP, subcortical plate; IZ, intermediate zone; SVZ, subventricular zone; VZ, ventricular zone; VZD, ventricular zone disruption; CC, corpus callosum; IC, internal capsule; GSC, glial stem cell; RGSC, reactive glial stem cell; GP, glial progenitor; RGP, reactive glial progenitor; A, astrocyte; RGC, radial glial cell; NP, neural progenitor; PVH, periventricular heterotopia.

## Data Availability

Not applicable.

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
