# Peer review of "AQP4, Astrogenesis, and Hydrocephalus: A New Neurological Perspective"

_ijms, 2022, doi:10.3390/ijms231810438_

Round 1
Reviewer 1 Report
The authors present a review of the current research landscape on AQP4 19
as a possible biomarker for gliogenesis and its influence in pediatric hydrocephalus. In fact, the manuscript is more an opinion piece/description of a hypothesis than a review. I think this should be mentioned more clearly in the manuscript. Indeed, the authors support their theory with targeted articles.
Even if not all children (by far) with hydrocephalus experience a clinical neurodevelopmental delay/impairment, the theory is interesting.
I have one request for the authors. At the end of section 2. they concluded that complementary treatments should not only focus on treating intracranial hypertension but also on preventing cellular neurodevelopmental impairments. Can the authors detail a little bit what are the potential treatments to prevent these cellular neurodevelopmental impairments?
Author Response
Leandro Castaneyra-Ruiz, PhD.
Sr. Scientist I
CHOC Children’s Research Institute
1201 W. La Veta Avenue
Orange, CA 92868
Dear Reviewer 1
We were pleased to receive the positive reviews of our manuscript entitled “AQP4, Astrogenesis, and Hydrocephalus; a new neurological perspective” from IJMS. We appreciate the Reviewers’ thoughtful comments and have worked to address each point in the attached revised version of the manuscript. Below, please find the response to the reviewer comments:
The authors present a review of the current research landscape on AQP4 19
as a possible biomarker for gliogenesis and its influence in pediatric hydrocephalus. In fact, the manuscript is more an opinion piece/description of a hypothesis than a review. I think this should be mentioned more clearly in the manuscript. Indeed, the authors support their theory with targeted articles.
Response: The hypothetical nature of AQP4 as a biomarker of astrogenesis and its relation with pediatric haydrocephalus has been included at the end of the introduction.
Even if not all children (by far) with hydrocephalus experience a clinical neurodevelopmental delay/impairment, the theory is interesting.
I have one request for the authors. At the end of section 2. they concluded that complementary treatments should not only focus on treating intracranial hypertension but also on preventing cellular neurodevelopmental impairments. Can the authors detail a little bit what are the potential treatments to prevent these cellular neurodevelopmental impairments?
Response: A paragraph at the end of point 2 has been included to detail the nature of the potential treatments that can prevent cellular neurodevelopmental impairments.
Reviewer 2 Report
This review is discussing AQP4, Astrogliogenesis, and Hydrocephalus; a new neurological perspective. Authors discussed about the AQP4 as a possible biomarker for gliogenesis and its influence in pediatric hydrocephalus, which would help to understand brain development under hydrocephalic and normal physiologic conditions. It is very interesting review included comprehensive information about AQP4 in relation to hydrocephalus. Just some issues needed to be considered:
1. In the section 1.2 authors mentioned “Pediatric hydrocephalus: a neurodevelopmental disorder.” Hydrocephalus in not neurodevelopmental disorder, it is one of the complications of the neurodevelopmental disorders or neurodevelopmental disorders are one of the complications of hydrocephalus. That needs to be changed.
2. What is the meaning of premature reactive astrogliosis? Premature reactive astrogliosis does not have the same meaning as differentiation of premature cells into the astrocytes. It needs to be revised.
3. Fig2. Legend needs more explanation and considering the same comment as comment 2.
4. Since the title of paper is included AQP4 and astrogliosis , it would be great if authors add another section about astrogliosis.
5. Since the main conclusion of review is suggesting AQP4 as a possible biomarker for gliogenesis, it is important to discuss about other gliosis markers and compare with AQP4 to show what are the reasons of suggesting AQP4 as a possible biomarker for gliogenesis
Geliogenesis is the common definition about all types of glia cells , it is important to consider astrogenesis since the focus of this review is on the AQP4 which is expressed mostly on the end-feet of astrocytes
Author Response
Leandro Castaneyra-Ruiz, PhD.
Sr. Scientist I
CHOC Children’s Research Institute
1201 W. La Veta Avenue
Orange, CA 92868
Dear reviewer 2
We were pleased to receive the positive reviews of our manuscript entitled “AQP4, Astrogenesis, and Hydrocephalus; a new neurological perspective” from IJMS. We appreciate the Reviewers’ thoughtful comments and have worked to address each point in the attached revised version of the manuscript. Below, please find the response to the reviewer comments:
This review is discussing AQP4, Astrogliogenesis, and Hydrocephalus; a new neurological perspective. Authors discussed about the AQP4 as a possible biomarker for gliogenesis and its influence in pediatric hydrocephalus, which would help to understand brain development under hydrocephalic and normal physiologic conditions. It is very interesting review included comprehensive information about AQP4 in relation to hydrocephalus. Just some issues needed to be considered:
- In the section 1.2 authors mentioned “Pediatric hydrocephalus: a neurodevelopmental disorder.” Hydrocephalus in not neurodevelopmental disorder, it is one of the complications of the neurodevelopmental disorders or neurodevelopmental disorders are one of the complications of hydrocephalus. That needs to be changed.
Response: the title of section 1.2 has been changed to “AQP4 in pediatric hydrocephalus: neurodevelopmental implications”.
- What is the meaning of premature reactive astrogliosis? Premature reactive astrogliosis does not have the same meaning as differentiation of premature cells into the astrocytes. It needs to be revised.
Response: “premature reactive astrogliosis” has been changed to “premature astrogenesis.”
- Fig2. Legend needs more explanation and considering the same comment as comment 2.
Response: The legend in figure 2 has been updated.
- Since the title of paper is included AQP4 and astrogliosis, it would be great if authors add another section about astrogliosis.
Response: The title is “AQP4, Astrogenesis, and Hydrocephalus; a new neurological perspective”. We do not think it is necessary to include a new section. However, if the reviewer requires it and the editorial extends our time to answer the reviewers, we are happy to include it.
- Since the main conclusion of review is suggesting AQP4 as a possible biomarker for gliogenesis, it is important to discuss about other gliosis markers and compare with AQP4 to show what are the reasons of suggesting AQP4 as a possible biomarker for gliogenesis
Gliogenesis is the common definition about all types of glia cells, it is important to consider astrogenesis since the focus of this review is on the AQP4 which is expressed mostly on the end-feet of astrocytes.
Response: Gliogenesis refers to the development of all glial cells, as pointed out by the reviewer, and our manuscript focuses on astrocytes. The term gliogenesis has been changed to astrogenesis. Thus, other astrogenesis markers have been included at the end of section 1.1 to compare with AQP4.
Round 2
Reviewer 2 Report
Authors responded to the comments in a satisfactory manner